# Ligand control of regioselectivity in palladium-catalyzed heteroannulation reactions of 1,3-Dienes

Dasha Rodina [1,2], Jakub Vaith [1,2] & Shauna M. Paradine [1] ✉

Olefin carbofunctionalization reactions are indispensable tools for constructing diverse, functionalized scaffolds from simple starting materials. However, achieving precise control over regioselectivity in intermolecular reactions remains a formidable challenge. Here, we demonstrate that using $PAd_2{}^nBu$ as a ligand enables regioselective heteroannulation of *o*-bromoanilines with branched 1,3-dienes through ligand control. This approach provides regiodivergent access to 3-substituted indolines, showcasing excellent regioselectivity and reactivity across a range of functionalized substrates. To gain further insights into the origin of selectivity control, we employ a data-driven strategy, developing a linear regression model using calculated parameters for phosphorus ligands. This model identifies four key parameters governing regioselectivity in this transformation, paving the way for future methodology development. Additionally, density functional theory calculations elucidate key selectivity-determining transition structures along the reaction pathway, corroborating our experimental observations and establishing a solid foundation for future advancements in regioselective olefin difunctionalization reactions.

Olefin carbofunctionalization reactions are powerful scaffold-building transformations that generate diversely functionalized products from simple starting materials[1,2]. These reactions form multiple new C–C and C–heteroatom bonds in a single step, with carbometallation of the olefin being a crucial elementary step that both enables further reactivity and dictates regioselectivity[3,4]. Despite decades of synthetic advances in carbofunctionalization reactions and other related coupling reactions of olefins, controlling regioselectivity of carbometallation in intermolecular reactions remains a significant challenge. Existing strategies for overriding innate substrate bias in reactions involving carbometallation of olefins have primarily relied on substrate control (e.g., directing groups, Fig. 1a)[5,6] or the use of stoichiometric additives (Fig. 1a)[7–13]. Catalyst-mediated control over regioselectivity of olefin addition presents an attractive alternative as it allows for fine-tuning of reaction outcomes, making it a potentially more generalizable approach.

Recently, we reported methods for Pd-catalyzed heteroannulation[14] of 1,3-dienes with *o*-bromoanilines and *o*-bromophenols using ureas as sterically-undemanding pro-ligands to overcome longstanding synthetic limitations in the heteroannulative synthesis of 2-substituted indolines[15] and dihydrobenzofurans[16]. During our ligand structure-reactivity investigations in the indoline synthesis, we observed trace formation of a by-product in our model reaction between *N*-tosyl *o*-bromoaniline and myrcene, which we later identified as the 3-substituted indoline. We hypothesized that this regioisomer was formed through a divergent 2,1-carbopalladation step, contrasting with the conventional 1,2-carbopalladation that leads to 2-substituted indoline formation (Fig. 1b)[17]. Our aim was to identify a ligand with the appropriate steric and electronic properties to selectively promote the 2,1-carbopalladation pathway and provide regiodivergent access to 3-substituted indolines[18]. To accomplish this, we needed to overcome the inherent electronic bias

[1]Department of Chemistry, University of Rochester, Rochester, NY, USA. [2]These authors contributed equally: Dasha Rodina, Jakub Vaith.
✉e-mail: sparadin@ur.rochester.edu

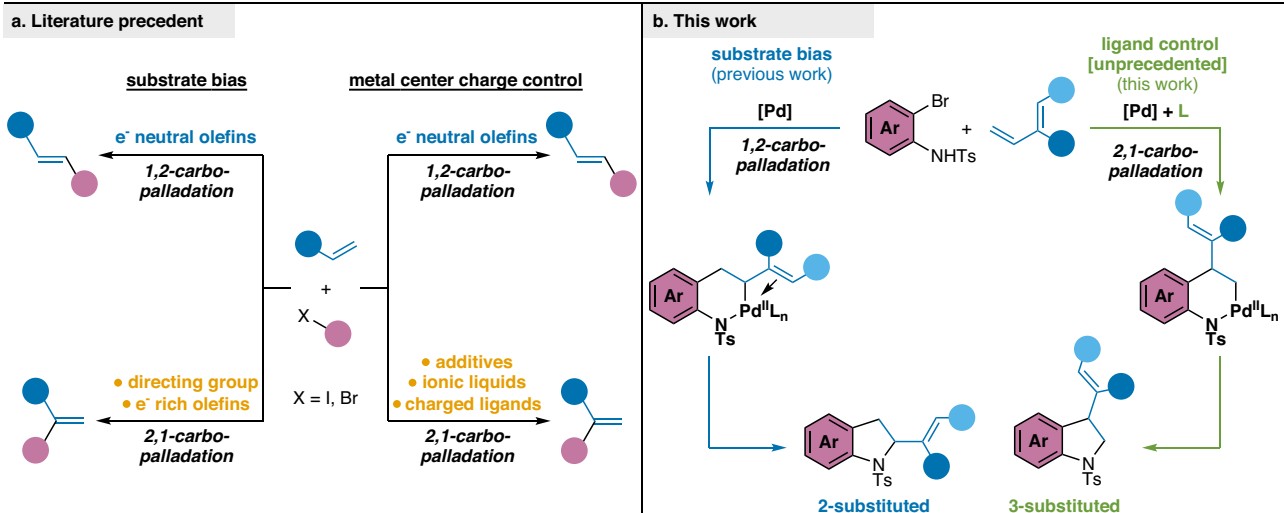

**Fig. 1 | Approaches to carbopalladation selectivity control. a** General strategies used to reverse carbopalladation to overcome substrate bias and achieve 2,1-carbopalladation include use of directing group on the substrate and electron rich (e⁻ rich) olefins, or metal center charge control by use of additives, ionic liquids or charged ligands. **b** This work: ligand-controlled, regioselective heteroannulation overrides substrate bias to provide access to complementary heterocycles. Ar = aryl, e⁻ = electron, L = ligand, Ts = *p*-toluenesulfonyl.

for 1,2-carbopalladation, which situates the Pd adjacent to the remaining olefin[19].

In this work, we demonstrate ligand control over regioselectivity in a 1,3-diene carbofunctionalization reaction. Using PAd₂ⁿBu (Cat-aCXium A, **L1**) as ligand, we have achieved highly regioselective heteroannulation of *o*-bromoanilines and branched 1,3-dienes, offering regiodivergent access to 3-substituted indolines and delivering excellent regioselectivity and good reactivity for a wide range of functionalized substrates. To further our understanding of selectivity control in this transformation, we employ a data-driven strategy by developing a linear regression model capable of predicting regioselectivity in the model reaction, using phosphorus ligand parameters from the Kraken database[20]. This predictive model highlights four crucial parameters that govern regioselectivity control in this transformation. Additionally, we use density functional theory (DFT) calculations to examine the selectivity-determining steps along the reaction pathway with a model 3-selective ligand (**L2**). The outcomes of these calculations corroborate our experimental observations; together, these studies substantiate our findings and have established a foundation for further methodological advancements in regiodivergent olefin difunctionalization reactions.

## Results and discussion
### Ligand structure-reactivity and selectivity studies
To investigate the ligand structure-reactivity and selectivity relationship, we used *N*-tosyl *o*-bromoaniline (**1a**) and myrcene (**2a**) as model substrates and examined the impact of various phosphine ligands on selectivity and reactivity in the heteroannulation reaction (Fig. 2a). We began with conditions optimized in our previous paper for urea ligands and the synthesis of 2-substituted indolines. Under these conditions, all initially tested phosphine ligands were inhibitory (yields <32%) compared to the reaction without exogenous ligand (32% yield) and yielded 2-substituted indoline as the primary product. However, with **L1**, we observed an enhancement of the combined yield to 56%, and more importantly, **3a** was the dominant product in a 92:8 regioisomeric ratio (r.r.) (**3a:4a**). We then optimized the initial reactivity and selectivity, allowing us to generate the desired product in high yield (88%) with excellent regioselectivity (>95:5 r.r.); this was accomplished by lowering the reaction temperature from 120 °C to 100 °C and switching the metal precursor from Pd(OAc)₂ to Pd₂(dba)₃. We

hypothesize that the increased regioselectivity is partly due to the reduction of background reactivity arising from phosphine-free [Pd] when starting with a Pd(II) salt.

Under these optimized conditions, we performed a two-round ligand screening, investigating the effects of 20 ligands in the first round and 22 additional ligands in the second round. The first-round ligands broadly covered the steric-electronic monodentate phosphorus ligand space (Fig. 2b). Without an exogenous ligand, the combined product yield in the model reaction was 21%, with the 2-substituted indoline product **4a** being strongly favored (9:91 r.r.). Even in this expanded screen, nearly all ligands were inhibitory compared to the reaction without an exogenous ligand, with **L1** and **L9** being the only ligands that led to a yield above 30%. While most first-round ligands still favored the 2-substituted indoline, twelve ligands (e.g., PⁱBu₂Me (**L2**), PCy₃, **L4**, Fig. 2b, c) shifted the regioselectivity towards the 3-substituted indoline (>30:70 r.r.). Using the ligand parameters from the Kraken database, we were unable to identify a single parameter that would predict selectivity inversion in the model reaction, though we did observe certain trends. Electron-deficient ligands did not invert regioselectivity. Regarding steric parameters such as pyramidalization P or minimum of percent buried volume (%V_bur(*min*)), the ligands that significantly inverted regioselectivity were neither small nor large (%V_bur(*min*) between 28 and 33)[21]. These trends informed our second round of ligand screening, which identified five additional ligands that inverted regioselectivity and provided sufficient data points to transition from qualitative to quantitative analysis.

### Ligand selectivity analysis through linear regression
To further explore the factors governing how ligand structure affected the regioselectivity on the model reaction, multivariate linear regression analysis was performed on our experimental data (Fig. 2b, see Supplementary Code 1 for algorithm design), using ligand parameters from the Kraken database. The lack of spread in our product yield data prevented us from performing an analogous analysis of ligand effects on yield. We converted the measured regioselectivity into $\Delta G^{\ddagger}$ using the Gibbs free energy equation ($\Delta\Delta G^{\ddagger} = -RT\ln(r.r.)$), to model the differential energy between the transition states of the two competing carbopalladation pathways (i.e., 1,2-carbopalladation vs. 2,1-carbopalladation). We organized our results into three classes based on the observed 3-selectivity (<30%, 30−75%, >75%), and each group's data

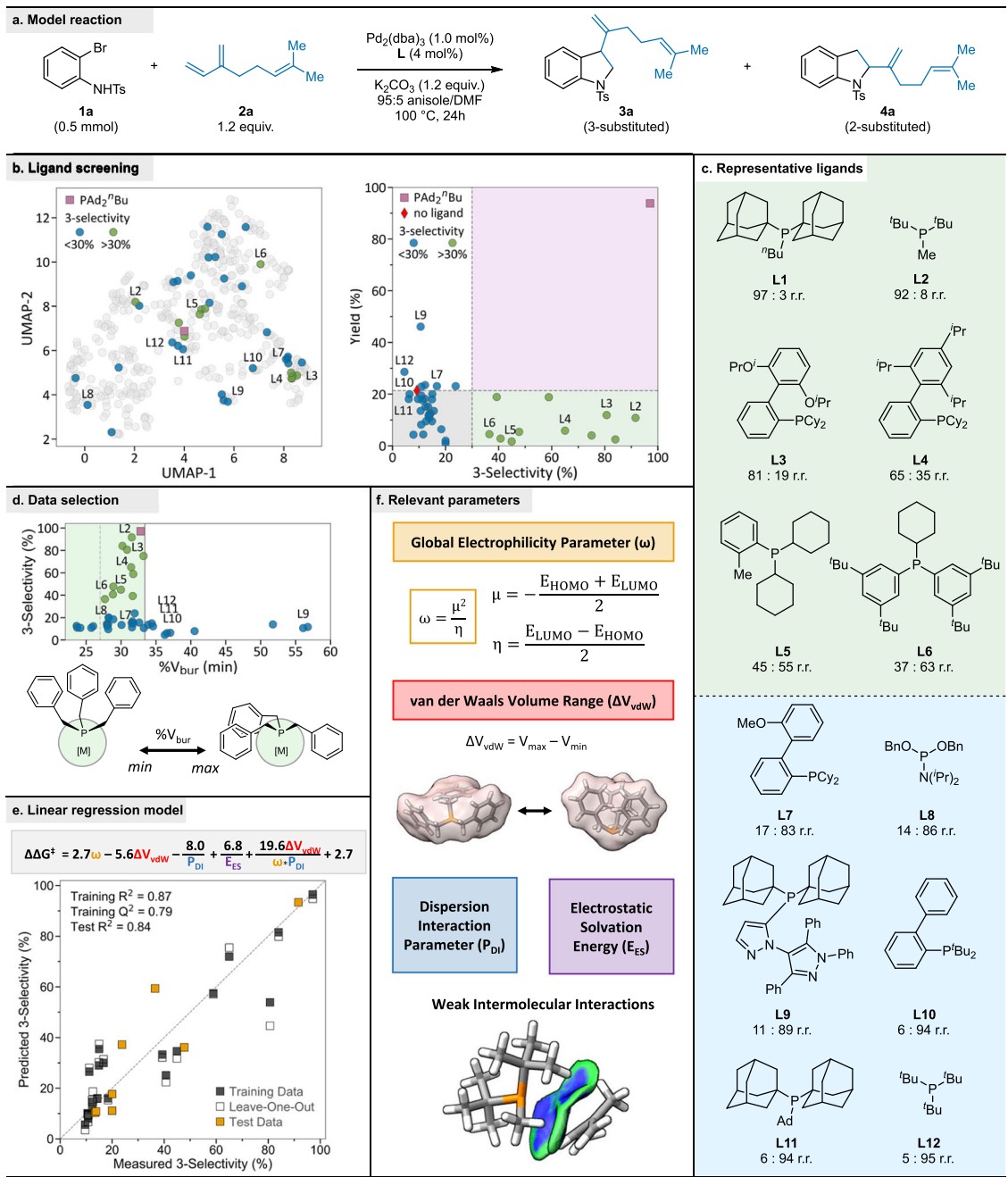

**Fig. 2 | Ligand structure-reactivity and selectivity studies. a** Model reaction and conditions used for the ligand screening. **b** 2D visualization of phosphorus ligand space and reaction selectivity results with yield. **c** Selected examples from the ligand screen with selectivity results. **d** Relationship between selectivity and percent buried volume of the ligands, highlighting the ligand cut off at $V_{bur}(min)$ 33%.

**e** Linear regression model reveals four key ligand parameters that influence selectivity. **f** Visual representations of key parameters. Ad = adamantyl, dba = dibenzylideneacetone, DMF = *N,N*-dimethylformamide, [M] = metal, r.r. = regioselectivity ratio, UMAP = uniform manifold approximation and projection for dimension reduction.

was randomly split into a training and validation set, following a 75:25 partition. We used an exhaustive-search linear regression algorithm to identify initial model candidates, which were evaluated based on their $R^2$ values and also $Q^2$ values from Leave-One-Out cross-validation (LOOCV) analysis.

Our first attempt was to apply this search to our entire data set, but we were unable to identify any viable model candidates. This led us to hypothesize the existence of two distinct mechanistic pathways within our data set, each with a different regioselectivity control mechanism. Our hypothesis was supported by the high 2-selectivity observed with the most sterically demanding ligands (e.g., **L10-12**) compared to

conditions without exogenous ligands and all other ligands. Specifically, we hypothesized that highly sterically-demanding ligands lead to the formation of an underligated, three-coordinate palladium complex in the regioselectivity determining step, while for the remaining ligands, a four-coordinate palladium species would be expected. Density functional theory (DFT) calculations that support our mechanistic hypothesis are presented in the next section of this paper.

To test our hypothesis, we identified the $\%V_{bur}(min)$ parameter as a potential criterion for separating ligands into two classes, as it has been identified as a good predictor of changes in phosphorus ligand ligation states[21]. Our selectivity data, plotted against this parameter

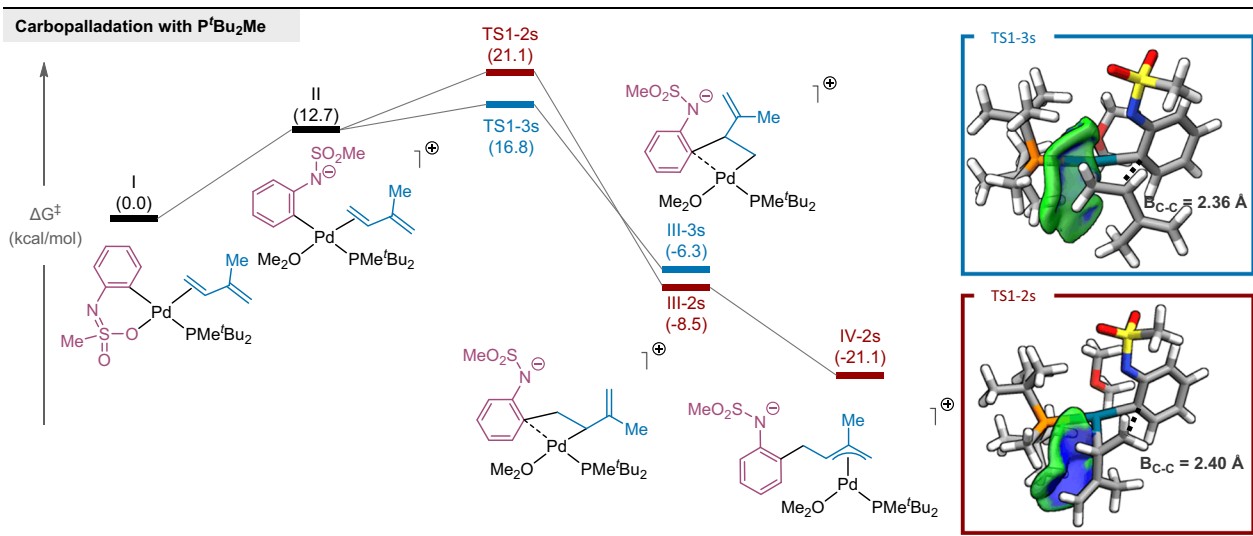

**Fig. 3 | Computational studies of the regioselectivity determining carbopalladation step.** Optimal pathway with **L2** as a model 3-selective ligand. DFT calculations were performed at the SMD(acetone)-ωB97X-D/6-311++G(2d,p)/SDD(Pd) level of theory. Values in parentheses are Gibbs free energies in kcal/mol. TS = transition state.

(Fig. 2d), clearly shows that no 3-selective ligands are found above % $V_{bur}(min)$ = 33. However, since the 3-selective ligands are all clustered within the intermediate range for % $V_{bur}(min)$, we decided to examine the impact of eliminating both the largest and smallest ligands (% $V_{bur}(min)$ >33 and <28, respectively) on the viability of potential model candidates. Removal of the smallest ligands did not yield any suitable model candidates, but when we excluded ligands with % $V_{bur}(min)$ > 33, consistent with our hypothesis, we were able to identify several suitable model candidates. The initial exhaustive search was used to identify four-term models dependent on three parameters and a cross-term. The final model (Fig. 2e) was then identified by testing addition of a fourth parameter to the selected model candidates. A five-term model using four parameters and a cross-term [coefficient of determination ($R^2$) = 0.87] was identified to describe the observed regioselectivity. Cross-validation techniques, including LOOCV [$Q^2$ = 0.79] and external validation [test $R^2$ = 0.84], were performed to indicate that a statistically robust model was produced.

Our model depends on the following parameters: (1) global electrophilicity parameter (ω), which describes the overall Lewis acidity of a molecule, (2) the van der Waals volume range of the ligand ($\Delta V_{vdW}$), which describes the range of intrinsic molecular volume, (3) the dispersion interaction parameter ($P_{DI}$), which describes the London dispersion ability of a molecule, and 4) the electrostatic component of solvation energy ($E_{ES}$), which describes the electrostatic interactions that occur in solution; in addition, there is a conformation-dispersion-electrophilicity interaction cross-term (Fig. 2f). Overall, the model emphasizes the importance of weak noncovalent interactions, such as electrostatic interactions and London dispersion effects, between the ligand and the reagent fragments. The role of these interactions as factors in regioselectivity control in the model heteroannulation was investigated further through computational analysis.

## Computational analysis of selectivity-determining steps

We turned to DFT calculations to examine the regioselectivity-determining steps along the reaction pathway and to gain insights into how the parameters from the linear regression model influence interactions between the ligand and the reagent fragments in the carbopalladation transition states, ultimately giving rise to regioselectivity in the heteroannulation reaction. For the computational models, we used mesyl-protected *o*-bromoaniline (**1a**-Ms), which still exhibited high regioselectivity in the reaction with **2a** (see Supplementary Information for more details), and isoprene (**2b**). Additionally,

we used the second most 3-selective ligand, P$^t$Bu$_2$Me (**L2**), which is less computationally demanding than our optimal ligand **L1**, but shares a similar structure. While this combination was suboptimal under our standard catalytic conditions, with a slight increase in catalyst loading and reaction temperature, we were able to isolate product **3ab**-Ms in 34% yield and >95:5 r.r.; this 3-selectivity is comparable to our experimental model reaction under nearly identical conditions. Since the focus of this work is on ligand control of regioselectivity in carbopalladation, we restricted our computational studies to the key migratory insertion/carbopalladation step, starting from the oxidative addition complex **I** after halide displacement (Fig. 3).

After assessing several potential reaction pathways, we identified the "trans-cationic" pathway to be the only one that accurately predicted the observed 3-selectivity in the experimental reaction (see Supplementary Information for analysis of other pathways). This pathway commences from the oxidative addition complex following halide displacement (therefore it is "cationic" and has the phosphine ligand and the aryl group in a trans configuration). The "cationic" pathway being operational in our reaction is supported by the inhibitory effect of halide anion addition (Table S1, entries 19-21), similar to what is observed in "cationic" Heck reactions[13,22].

The 2,1- and 1,2-carbopalladation steps in the "trans-cationic" pathway are both moderately exergonic, with the 2-selective σ-complex **III-2s** being slightly favored by 2.2 kcal/mol. While isomerization of σ-complex **III-2s** into the π-allyl complex **IV-2s** provides significant additional stabilization (−12.6 kcal/mol), no such rearrangement is possible for σ-complex **III-3s**. Consequently, the carbopalladation step is likely irreversible, suggesting that product selectivity is under kinetic control. In a computational study of their regioselective Heck arylation, the Mecking group determined that the cis arrangement of the ligand and diene is essential for effective control of carbopalladation regioselectivity via ligand-alkene steric interactions[8]. The "trans-cationic" pathway, identified as the probable pathway in our reaction, features the phosphine ligand and diene in this beneficial cis configuration, optimally positioning them for steric interactions that can influence regioselectivity. The 3-selective transition state, **TS1-3s**, exhibits weak repulsive ligand-diene interactions and is significantly favored over **TS1-2s** (−4.2 kcal/mol, >99:1 r.r. at 100 °C), which has stronger repulsive interactions, as evaluated using an independent gradient model analysis[23]. These results are consistent with the observed 3-selectivity in our model reaction with **L2** and **1a**-Ms (>95:5 r.r.). These interactions destabilize the conventionally favored

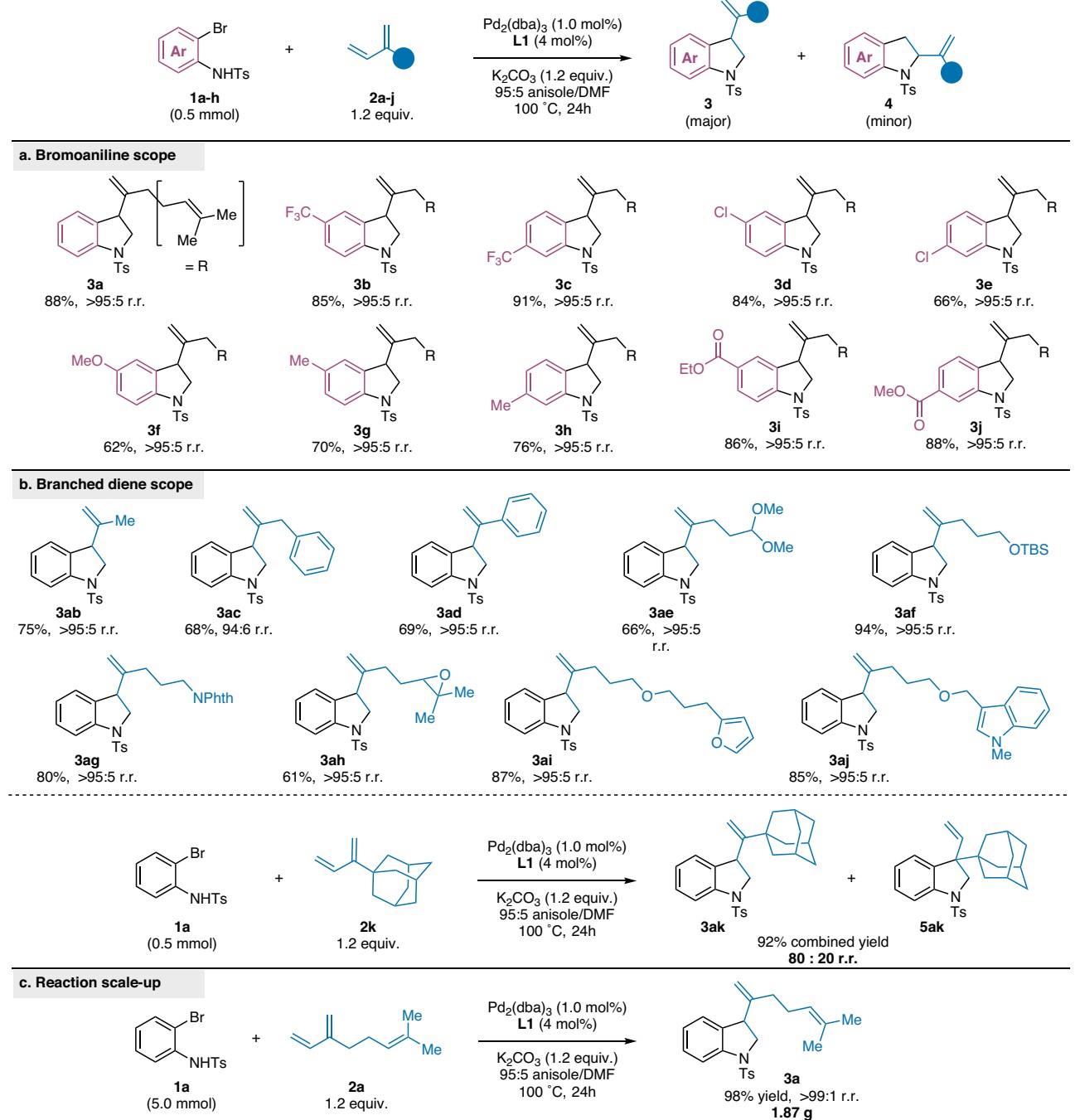

**Fig. 4 | Substrate Scope of *o*-bromoanilines and 1,3-dienes.** Standard reaction conditions: **1** (0.5 mmol), **2** (0.6 mmol), Pd₂(dba)₃ (0.005 mmol), **L1** (0.02 mmol), K₂CO₃ (0.6 mmol), 95:5 anisole/DMF (0.5 M), 100 °C. **a** Scope of N-tosyl-1,2-bromoanilines. **b** Scope of branched 1,3-dienes. **c** Gram-scale reaction. NPhth = phthalimide, TBS = *tert*-butyldimethylsilyl.

1,2-carbopalladation transition state, thus overriding the inherent substrate bias and allowing for regioselective synthesis of 3-substituted indolines, and are accounted for in our quantitative linear regression model through the dispersion interaction parameter ($P_{DI}$), the electrostatic component of solvation energy ($E_{ES}$), and the conformation-dispersion-electrophilicity interaction cross-term. It is expected that the conformational flexibility of the ligand, approximated in the model through the van der Waals volume range ($\Delta V_{vdW}$), would also impact the strength of these interactions. While we cannot currently rule out a scenario wherein carbopalladation vs. aminopalladation is selectivity-determining, our calculations and experimental observations (e.g., reversion of regioselectivity with phenyl-conjugated diene **2l**,

see Fig. 5a) are consistent with our mechanistic hypothesis of 1,2- vs. 2,1-carbopalladation being selectivity-determining.

## Substrate scope

We explored the generality of our method with various *o*-bromoanilines (Fig. 4a) and branched 1,3-dienes (Fig. 4b). Both electron-rich and electron deficient bromoanilines (**3b-3h**) were good substrates under our reaction conditions, producing 3-substituted indolines in 62% to 91% yields with excellent regioselectivity (>95:5 r.r.). Our method was also effective with ester-containing substrates (**3i**, 86% yield and **3j**, 88% yield) that closely resemble bioactive alkaloids such as benzastatins E and G[24].

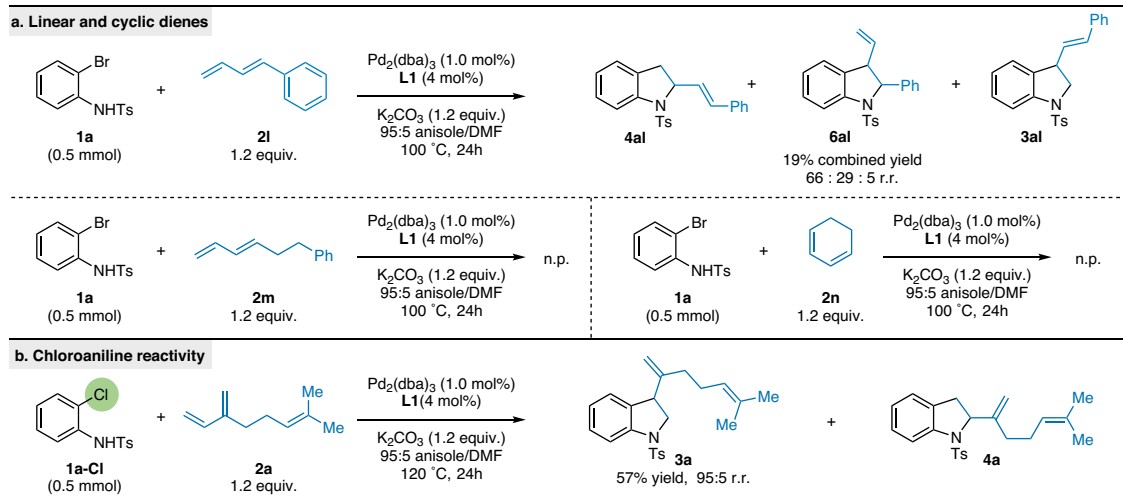

**Fig. 5 | Additional Substrates Probing Reaction Tolerance. a** Method limitations: linear and cyclic dienes. **b** Heteroannulation reactivity with chloroanilines. n.p. no product.

Our method accommodates a diverse scope of branched dienes. Both alkyl-substituted dienes (**3ab** and **3ac**) and aryl-substituted dienes (**3ad**) afforded 3-substituted indolines in good yield and excellent regioselectivity (>95:5 r.r.). Furthermore, 3-substituted indolines containing dimethyl acetal **3ae** (66%), silyl-protected alcohol **3af** (94%), and phthalimide-protected amine **3ag** (80%) were readily obtained under our reaction conditions. Branched dienes bearing heterocycles such as epoxides, furans, and indoles, were also suitable substrates (**3ah**-**3aj**), While examining the tolerance of our system to the steric properties of the diene substrate, we observed excellent yields with adamantyl diene **2k**, yet different selectivity compared to other branched dienes (80:20 r.r.). Interestingly, the minor product **5ak** resulted from the annulation of the internal alkene rather than 1,2-carbopalladation of the terminal alkene. The heteroannulation reaction is readily scaled up 10-fold to afford **3a** in quantitative yields and excellent regioselectivity (Fig. 4c).

We expanded our investigation to include other related substrates. First, we explored other diene classes (Fig. 5a). When we tested a commonly used linear conjugated diene (**2l**), the selectivity reverted to 2-substituted indoline (**4al**) as the major product and the yield diminished (19% combined yield). Additionally, linear diene **2 m** as well as cyclohexadiene (**2n**) yielded no heteroannulation product. Finally, we turned our focus on the ambiphile. By increasing the reaction temperature to 120 °C, we successfully engaged *o*-chloroaniline **1a-Cl** with myrcene **2a**, obtaining indoline **3a** in moderate yield (57%) and high regioselectivity (Fig. 5b). To our knowledge, this represents the first instance of successful heteroannulation of *o*-chloroanilines with dienes[25], demonstrating the potential for expanding the scope of this synthetic approach to include a wider range of substrate classes.

Through the exertion of ligand control over carbopalladation, we have successfully achieved a highly regioselective heteroannulation reaction of *o*-bromoanilines and branched 1,3-dienes. This strategy enables the efficient synthesis of 3-substituted indolines and displays exceptional selectivity and reactivity across a diverse range of functionalized substrates. Furthermore, our data-driven approach that employs a linear regression model, using phosphorus ligand parameters from the Kraken database, has provided valuable insights into the key parameters governing selectivity in this transformation. Our findings have highlighted the key role of weak noncovalent interactions between the ligand surface and substrates in influencing reaction outcomes. Additionally, DFT calculations have corroborated our experimental observations and shed light on the selectivity-determining steps in the catalytic cycle. We anticipate that the demonstrated application

of ligand control over carbometallation in palladium catalysis, along with predictive models for reaction selectivity, will have a transformative impact on the field of olefin carbofunctionalization.

## Methods
### General procedure for ligand-controlled heteroannulative synthesis of 3-substituted indolines
*N*-tosyl-bromoaniline **1a** (163 mg, 0.500 mmol, 1.0 equiv.), myrcene **2a** (103 μL, 0.600 mmol, 1.2 equiv.), di(1-adamantyl)-*n*-butylphosphine **L1** (7.2 mg, 0.020 mmol, 0.04 equiv.), potassium carbonate (83 mg, 0.600 mmol, 1.2 equiv.), and tris(dibenzylideneacetone)dipalladium(0) (4.6 mg, 0.005 mmol, 0.010 equiv.) were weighed out in the above-mentioned order into a 1-dram vial equipped with a stir bar and a cap with a silicone septum. The vial was then placed under a nitrogen atmosphere and charged with 1 mL of freshly degassed anisole/dimethylformamide (95:5) solvent mixture. The reaction was stirred at 100 °C for 24 h. After cooling to room temperature, regioisomeric ratios were determined by HPLC analysis of the crude reaction mixture and the reaction mixture was filtered with ethyl acetate through cotton plug. The solvents were removed under reduced pressure and the crude mixture was purified by flash column chromatography to obtain products **3**.

## Data availability
Crystallographic data are available free of charge from the Cambridge Crystallographic Data Centre under reference CCDC 2263163. Source data are provided with this paper. Additional experimental details, experimental procedures, crystal structures, compound characterization, NMR spectra for all new compounds, ligand data sets, and computational details are available in the Supplementary Information. Data can also be obtained from the authors upon request. Source data are provided with this paper.

## Code availability
The Python code developed for the algorithm to identify the linear regression model is provided with this paper as part of the Supplementary Information.

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

## Acknowledgements

S.M.P. thanks the University of Rochester and the National Science Foundation (CHE 22-38081) for financial support. D.R. is supported by the NIH T32 CBI Training Program (T32-GM118283), and J.V. is supported by Bristol-Myers-Squibb (Graduate Fellowship in Synthetic Organic Chemistry). We thank Amanda Canfield for checking the experimental procedure for the preparation of compound **3j** in Fig. 4a, Dr. William Brennessel for conducting X-ray crystallography, and Natalie Seeger and Prof. Matthew Sigman for their assistance with the Kraken database.

## Author contributions

D.R. and J.V. conducted all the experimental work, analyzed the data, and completed the DFT calculations. J.V. developed the Python script and performed the regression analysis. J.V. conceived the project, and D.R., J.V., and S.M.P. designed the project and wrote the manuscript. S.M.P. directed the research.

## Competing interests

The authors declare no competing interests.
