## [Peer Review File · Nature Communications]

REVIEWER COMMENTS

Reviewer #1 (Remarks to the Author):

In this work Paradine and co-authors reported a new strategy to access 3-substituted indoline products from 1,3-dienes and ambiphilic molecules. It is exciting to see that the regioselectivity can be controlled by ligands, thus complementing the existing methods, which will usually lead to 2-substituted indolines. They explained the origin of the regioselectivity in two ways: 1) They showed how the key parameters would affect the reaction; 2) They delivered DFT calculations to support the 2,1-migratory insertion mechanism. Given the fact that annulation between alkenes and ambiphiles remain popular during the past few decades, we could foresee that this work will inspire more regioselective annulations by ligand design. Although I have some concerns regarding the model they use and the scope of this work is somewhat limited, I lean to the acceptance of this work after the following concerns have been addressed.

(1) The modeling formula of Gibbs free energy in figure 2e is very confusing without explanation of each term. Could you please explain the meaning of each term in your model? For example, why you use a cross-term of electrophilicity parameter, etc. You should use linear regression to fit the coefficient of the variable only after you have a correct model.

(2) In this work the authors proposed a 2,1-carbopalladation pathway to explain the regioselectivity. This looks like a reasonable pathway based on their DFT calculation. However, I wonder if they have considered the Wacker-type aminopalladation pathway? See *Inorg. Chem.* 2007, 46, 6, 1910–1923 and Engle's recent annulation works.

(3) Figure 1a should be improved: They kept drawing the same product and saying "2,1-migratory insertion" again and again. They may want to save that space for more useful information.

(4) Do they have any explanations why the 3-substitution is required in the alkene? Have they ever tried any 3,4-disubstituted acyclic alkenes in the scope?

(5) Have they tried any other coupling partners such as O-/C-nucleophiles and anilines protected by other groups? Is it the yield or the selectivity issues?

(6) On page 3 they said “twelve ligands shifted the selectivity towards the 3- substituted indoline”. Based on figure 2c, there are only 6 ligands.

(7) ¹⁹F NMR should be added for the fluorine containing compounds.

(8) The ¹³C NMR for compound 1k (Page S4): where is the quartet CF₃ peak? A careful proofreading of the SI is suggested.

Reviewer #2 (Remarks to the Author):

Comments: This paper reported the ligand control strategy to synthesize 3-substituted indolines by Palladium-catalyzed heteroannulation reaction of o-bromoanilines and branched 1,3-dienes. This article combined the Kraken database, using multiple linear regression to show the origin of regioselectivity when different monophosphine ligands were used. We are very glad to see that the author's screening of phosphine ligand was rigorous and efficient, and this well-distributed sampling and experiment are very powerful for subsequent analysis. This model suggested that the weak noncovalent interactions should also be considered in the screening of phosphine ligands. According to the steric hindrance of ligands, different modes of active intermediates were proposed and verified by DFT calculations (three-coordinate palladium species for highly sterically-demanding monophosphine ligands and remaining monophosphine ligands). In the substrate scope, only 2-substituted 1,3-dienes showed high yield and selectivity. The tolerance of other functional groups in this reaction cannot be demonstrated through this limited scope. In general, this paper is rigorous and novel in its interpretation of reaction selectivity, and we expect that this deep understanding of the mechanism will further expand the application of this reaction. But a major revision is needed

1. More groups should be added to the substrates to the reaction scope (such as carbonyl, ester, halogen, etc.).

2. Can the linear regression model predict a new monophosphine that could further expand the reaction scope? We hope that this model can guide ligand screening or ligand designing.

3. The optimal reaction conditions of L1 were used as reaction conditions to test the reaction selectivity of other phosphines. But this could be wrong, especially in the case of weak noncovalent interactions emphasized in this paper. In this case, solvents may be an important influencing factor to that. Therefore, this should be tested.

Reviewer #3 (Remarks to the Author):

The paper by Rodina et al. describes the development of a regioselective palladium-catalyzed annulative coupling reaction to form 3-substituted indolines guided by multivariate regression of numerical ligand descriptors against selectivity and density functional theory (DFT) calculations. The authors' findings provide an attractive method to obtain these harder-to-access indolines and provide insight on the factors that influence the success of the reaction.

The paper is well written and the figures are visually appealing and informative. The Supplementary Information (SI) appears thorough and the repeated measurement of yield for each reaction is commendable.

There are a few places where the writing could be refined, as indicated in the attached file.

Reviewer 1

1. The modeling formula of Gibbs free energy in figure 2e is very confusing without explanation of each term. Could you please explain the meaning of each term in your model? For example, why you use a cross-term of electrophilicity parameter, etc. You should use linear regression to fit the coefficient of the variable only after you have a correct model.

See figure 2f for a visual representation of each parameter.

We have modified the text to provide a description of the identified parameters:

Our model depends on the following parameters: 1) global electrophilicity parameter (ω), which describes the overall Lewis acidity of a molecule, 2) the van der Waals volume range of the ligand (ΔV_{vdw}), which describes the range of intrinsic molecular volume, 3) the dispersion interaction parameter (P_D), which describes the London dispersion ability of a molecule, and 4) the electrostatic component of solvation energy (E_{ES}), which describes the electrostatic interactions that occur in solution; in addition, there is a conformation-dispersion-electrophilicity interaction cross-term (Figure 2f).

We chose our model based on exhaustive-search linear regression algorithm, which refers to a method of fitting a linear regression model by considering all possible subsets of predictor variables and selecting the best subset based on a chosen criterion, often maximizing model performance or minimizing an error metric. This is a common approach to linear regression modeling (e.g., *Chem. Sci.*, **2018**, 9, 2398-2412). We arrived at the model in figure 2e by following this approach, and when the addition of a new variable stopped having an improved effect on the model we decided to incorporate a cross-term. This significantly improved our model.

2. In this work the authors proposed a 2,1-carbopalladation pathway to explain the regioselectivity. This looks like a reasonable pathway based on their DFT calculation. However, I wonder if they have considered the Wacker-type aminopalladation pathway? See *Inorg. Chem.* **2007, 46, 6, 1910–1923 and Engle’s recent annulation works.**

Thank you for your comment, we have considered the possibility of aminopalladation as a potential pathway. As of now we are not able to completely rule out the aminopalladation pathway. However, as you noted, our DFT studies support 2,1-carbopalladation. Additionally, the reaction outcome is highly dependent on the substitution pattern on the diene, which would not be expected if C–C vs. C–N coupling were the selectivity-determining event. For example, for 4-phenyl-1,3-butadiene, we find that regioselectivity reverts to the thermodynamically favored 1,2-carbopalladation. It’s important to note that this substrate has a strong electronic bias for placing Pd at the internal position due to conjugation into the aromatic ring, while also lacking the branching substitution that our model suggests is key for overriding substrate bias via ligand-substrate interactions. Considering this and our calculations we lean toward 1,2- vs. 2,1-carbopalladation being selectivity-determining.

3. Figure 1a should be improved: They kept drawing the same product and saying “2,1-migratory insertion” again and again. They may want to save that space for more useful information.

Although there was no additional information we wished to convey through this figure, we have modified Figure 1a in order to simplify the illustration and improve the symmetry of our visuals.

4. Do they have any explanations why the 3-substitution is required in the alkene? Have they ever tried any 3,4-disubstituted acyclic alkenes in the scope?

We are interested in finding an explanation on exactly why the branched pattern is important for reactivity. Currently our model highlights the importance of non-covalent interactions between the substrate and ligand. More investigations are on the way.

We tried to use (E)-2-methyl-1-phenyl-1,3-butadiene as a substrate, but the reactivity was diminished, giving only trace amounts of annulated product. This substrate has been added to the SI (S33).

5. Have they tried any other coupling partners such as O-/C-nucleophiles and anilines protected by other groups? Is it the yield or the selectivity issues?

We have tried the reaction and its optimization with 2-bromophenol but reactivity was low under our standard reaction conditions. We intend to pursue these substrates further, but this is beyond the scope of the current work.

For our results with differently protected bromoanilines, see SI substrates **3a**-**Ms** and **3k** (S25).

6. On page 3 they said “twelve ligands shifted the selectivity towards the 3- substituted indoline”. Based on figure 2c, there are only 6 ligands.

We apologize for the confusion in the figure. We were not able to include the structure of every ligand due to space constraints, that is why we titled the section “Representative ligands” (which means not all of them, but selected examples). Note that the caption for Figure 2c says “selected examples,” showing that the structures drawn do not represent every ligand tested. If you look at figure 2b you can see the green dots that represent every ligand with shifted 3-selectivity. We have changed the figure reference in the text to figure 2b-c to avoid confusion.

7. ¹⁹F NMR should be added for the fluorine containing compounds.

¹⁹F NMRs were added to the SI for compounds **1k**, **3b**, **3c**, and **3k**.

8. The ¹³C NMR for compound 1k (Page S4): where is the quartet CF₃ peak? A careful proofreading of the SI is suggested.

Thank you for your suggestion. New ¹³C and ¹⁹F NMRs of this compound were added to the SI, and the C-F coupling is noted in the tabulated data for compound **1k** (page S4). Note that this data was also added for product **3k** (page S26), though not specifically requested by the reviewer.

Reviewer 2:

1. More groups should be added to the substrates to the reaction scope (such as carbonyl, ester, halogen, etc.).

See examples of substrates with carbonyl groups **3i** and **3j**, see substrates **3d** and **3e** for examples of halogens; these can be found in Figure 4 in the manuscript.

Additionally, see the SI for substrates that were tested but were unreactive under our reaction conditions (S33).

2. Can the linear regression model predict a new monophosphine that could further expand the reaction scope? We hope that this model can guide ligand screening or ligand designing.

The linear regression model was developed using branched dienes as substrates, and for this our model predicts **L1** to be the most selective ligand (at 98:2, it would be harder to get more selective). In order to predict ligands that would be more selective for different substrates (e.g., linear dienes, 2-bromophenols), we expect that additional ligand screening and development of a new linear regression model would be needed, as the ligand-substrate interactions identified as key to selectivity would be

different, which would likely lead to different ligand parameters being relevant for selectivity. This is certainly an avenue we intend to pursue, but is beyond the scope of the current work.

3. The optimal reaction conditions of L1 were used as reaction conditions to test the reaction selectivity of other phosphines. But this could be wrong, especially in the case of weak noncovalent interactions emphasized in this paper. In this case, solvents may be an important influencing factor to that. Therefore, this should be tested.

Thank you for your suggestion. Our Supplementary Information includes the solvent screens we conducted during our reaction development (see SI, S10-S11). In these screens, we found that solvent had minimal impact on the product selectivity using **L1**, despite a broad range of solvent polarity (toluene to DMSO). The only significant impact of solvent was on product yield. This observation led us to be confident in conducting the ligand analysis using the optimal reaction conditions for **L1**.

Reviewer 3:

1. The interpretation of the DFT calculations as presented is reasonable and supports the conclusions drawn from the linear regression. However, there are discrepancies between the experimental conditions, DFT parameters, and calculated structures. The optimized reaction was performed in a mixture of DMF/anisole. The caption of Figure 3 and the SI indicate that solvation parameters of acetone were employed. Figure 3 shows dimethyl ether as a coordinating solvent. Can the authors comment on the validity of their calculations given the substantial electronic and steric differences between DMF/anisole, acetone, and dimethyl ether?

In our solvent studies we see minimal effect on product selectivity, despite significant variation in solvent polarity (See SI, S10-S11). There is no exact solvation model for our reaction solvent mixture, so we chose acetone, as we had previously used for calculations in our first report of heteroannulation using urea ligands, that allowed for calculation comparison (*J. Am. Chem. Soc.* **2022**, *144*, 6667–6673). Dimethyl ether was used as a simplified version of anisole to relieve the computational burden, as the calculations were already quite computational complex. Given that solvent itself does not seem to affect product selectivity, we felt comfortable making this deviation from our experimental reaction conditions.

2. Differences in calculated energies are referred to as “exothermic” or “endothermic.” Should they not be referred to as “exergonic” or “endergonic” since Gibbs free energy is referenced?

Thank you for catching this. We have changed the terms in the paper to “exergonic” or “endergonic.”

3. Do the authors have any hypotheses as to why linear alkenes afford little to no product? I suspect that the geometry of the internal alkene may play a role, but I am curious if the authors elucidate this experimentally?

We hypothesize that electronic properties of the unreacted alkene are responsible for the inability of linear dienes engage in 2,1-carbopalladation. In our previous study we see that phosphine ligands are inhibitory to 1,2-carbopalladation pathway of this reaction (*J. Am. Chem. Soc.* **2022**, *144*, 6667–6673). We found that selective ligands can enable 2,1-carbopalladation but only for branched dienes. In the case of phenylbutadienes, i.e., (*E*)-1-Phenyl-1,3-butadiene, we see that the reaction reverts to the formation of small amounts of 2-substituted product and no 3-substituted product, which is consistent with phosphine ligands inhibiting the 1,2-carbopalladation. More mechanistic studies are needed to clarify what exactly is responsible for inhibition of 2,1-carbopalladation in linear dienes.

4. Figure 1 provides a good graphical summary of literature precedent. I suggest adding numbers on the olefin carbon atoms (i.e., 1 and 2) and the resultant intermediates to make the relationship between starting material and products clearer.

Thank you for your suggestion, part a of the figure was revised. In part b we use color to make the distinction between two starting materials, and new bonds that were made in intermediate steps.

5. Paragraph 2 of the Introduction references compounds 1a and 2a, but the structures are not shown until Figure 2. Figure 1b needs to be re-drawn to include these compounds.

We removed **1a** and **2a** from the introduction paragraph.

6. In paragraph 2 of the Introduction, the last sentence could be re-written. The location of the “allylic position” is not clear since there is no reference to a structure. Using atom numbers to reference the position might be more informative.

Sentence was revised to “...which situates the Pd adjacent to the remaining olefin.”

7. The topmost left arrow in Figure 1a is colored blue instead of black.

Figure 1a was revised.

8. Compound numbers could also be added to the starting material and products in Figure 1b to make it clearer which species the authors are referring to.

Figure 1 is an introductory figure and is more conceptual therefore we chose not to include compound numbers; this seems to be standard practice in the field.

9. Paragraph 1 of the Results and Discussion uses the symbol “[Pd]”. Figure 2a does not use this symbol. The Scheme is drawn specifically, whereas this paragraph discusses reaction optimization. Either the Scheme should be revised to indicate which parameters were varied, or the comments on optimization should be removed.

While we specifically referred to the Pd salts that we used in our reaction development in the text, we appreciate the confusion [Pd] may cause. We removed [Pd] from the text.

10. In paragraph 2 of the Results and Discussion there are a few awkward sentences: “Electron-deficient ligands did not invert selectivity, and regarding steric parameters...” could be reworded. Similarly, the statement “(%V_{bur}(min) between 28 and 33)” could be revised.

The sentence was split into two: “Electron-deficient ligands did not invert selectivity. Regarding steric parameters such as pyramidalization P or minimum of percent buried volume (%V_{bur}(min)), the ligands that significantly inverted selectivity were neither small nor large (%V_{bur}(min) between 28 and 33).

11. A very minor suggestion for Figure 2a: the desired 3-substituted product could be highlighted either with a text or box to increase clarity.

We have opted against circling the product, referring to it within the text by its designated identifier, **3a**. We did indicate which product was major and which was minor in order to increase clarity.

12. It is not clear what a “comparable analysis of ligand effects on yield” means.

The sentence was changed to “The lack of spread in our product yield data prevented us from performing an analogous analysis of ligand effects on yield.” We were trying to indicate that we are unable to use linear regression to derive ligand parameters responsible for influencing product yields.

13. The caption of Figure 3 contains the word “ligand” a few too many times and should be revised.

Thank you for catching this! The caption was revised.

REVIEWER COMMENTS

Reviewer #1 (Remarks to the Author):

The revised manuscript has properly resolved my concerns. I feel that this work is ready to be accepted.

Reviewer #2 (Remarks to the Author):

After reading the manuscript, I think that a major revision is still needed.

1, the computed regiochemistry is too high compared to experimental results. The difference in activation free energies for the regio-determining transition states are too high (more than 7 kcal/mol) and cannot explain the experimental results.

2, Calculations should use real ligands, not the model ligands.

3, Analysis of the regiochemistry is very misleading. The linear aggregation analysis gave little information. Based on the results, steric effect is critical. Analysis of the key steric interaction is more straightforward for chemists to understand their experimental results.

4, Global nucleophilicity is not a good parameter. Chemists are now using HOMO for nucleophilicity and LUMO for electrophilicity. See JOC, 2016,81, 5370.

5, The authors named the last figure in the manuscript wrongly as Figure 3, which should be Figure 4 exactly.

Reviewer 2

1. The computed regiochemistry is too high compared to experimental results. The difference in activation free energies for the regio-determining transition states are too high (more than 7 kcal/mol) and cannot explain the experimental results.

Thank you for your observation regarding the computed regiochemistry and its comparison with experimental results. Your point about the discrepancy in activation free energies is well taken. In our study, we modeled an alternative mechanism using underligated palladium species for a highly sterically demanding ligand; we had excluded such ligands from our linear regression model as it was clear that they were proceeding through a distinct pathway. It is important to clarify that this modeled mechanism does not represent an active intermediate in our optimized reaction. Our reaction employs a smaller ligand, proceeding through a 2,1-carbopalladation pathway.

While our computational results exhibit trends that are consistent with experimental observations, indicating that 1,2-carbopalladation is favored, we acknowledge that the energy difference observed is significantly higher than expected. This indicates that our computational model likely does not fully capture the nuances of the experimental system. In light of your feedback, we have reassessed this section of our work. We concur that these calculations, in their current form, are more preliminary than we initially considered.

Therefore, we have decided to move this section to the Supplementary Information of our paper. This decision is made to ensure that the main focus of the paper remains on the more robust and directly relevant findings. We believe that including these preliminary findings in the SI will provide interested readers with valuable insights into potential competing pathways, while acknowledging their preliminary nature.

2. Calculations should use real ligands, not the model ligands.

The ligand we employed for the calculations is indeed a real ligand that was experimentally tested under our reaction conditions. This ligand (L2), as mentioned in Figure 2 and further detailed in the SI, was selected based on its performance in triplicated experiments. It was found to be the most 3-selective ligand after the optimal ligand (L1) used for our substrate scope. It is important to note that although this particular ligand was not used in our scope investigations, its influence on the reaction is significant. The rationale behind examining the effects of P^tBu_2Me (L2) is to provide a comprehensive understanding of the reaction mechanism. This approach is justified by our regression analysis, which indicates that the impact of P^tBu_2Me (L2) on regioselectivity is comparably influential to that of $PA_{d_2}^tBu$ (L1). Both ligands notably alter the regioselectivity in an unprecedented manner.

3. Analysis of the regiochemistry is very misleading. The linear aggregation analysis gave little information. Based on the results, steric effect is critical. Analysis of the key steric interaction is more straightforward for chemists to understand their experimental results.

Regarding the linear regression analysis, we understand your point about its perceived limitations. However, this method was instrumental in identifying key parameters that significantly account for the observed regioselectivity among the 25+ ligands we studied. These parameters were identified with robust statistical backing and are pivotal for future ligand design. We would like to emphasize the value of this analysis in contributing to a deeper understanding of the regioselectivity phenomena, which is not immediately apparent but is crucial for the advancement of the field.

You point out the critical role of steric effects, however, our findings suggest that steric effects, while important, are not the sole contributors to the observed regioselectivity. This complexity is precisely why a single parameter fails to adequately predict regioselectivity. Our study demonstrates that regioselectivity is influenced by a combination of factors, necessitating a more comprehensive analysis. Regarding your suggestion for a more straightforward analysis of key steric interactions, we agree that simplicity in interpretation is desirable. However, in this particular case, the complexity of the chemical system under study precludes a simplistic approach. The linear regression analysis we employed was not an arbitrary choice but a necessary tool to develop a predictive model. This model captures the nuances of the system that a straightforward analysis might overlook. It is this depth and complexity of analysis that provides the value in our study, even though it may seem less direct.

4. Global nucleophilicity is not a good parameter. Chemists are now using HOMO for nucleophilicity and LUMO for electrophilicity. See JOC, 2016,81, 5370.

We think the reviewer is referring to global electrophilicity, which is one of the parameters included in our model. We have indeed considered the roles of both E_{HOMO} and E_{LUMO} , as illustrated in our paper, since global electrophilicity depends on both E_{HOMO} and E_{LUMO} (See Figure 2f). The Kraken database, which we utilized for our analysis, offers a diverse array of parameters, including global electrophilicity, E_{HOMO} , and E_{LUMO} . These parameters are interconnected in a non-linear manner, which is why they are valuable for our linear regression analysis. The non-linear interrelation between these parameters means that a simple linear analysis using just E_{HOMO} or E_{LUMO} might not fully capture the subtleties of our reaction's regioselectivity.

5. The authors named the last figure in the manuscript wrongly as Figure 3, which should be Figure 4 exactly.

Thank you for catching this. We changed it in the manuscript.

REVIEWER COMMENTS

Reviewer #4 (Remarks to the Author):

After reading the manuscript and the responses of the authors, I think the authors still need to use Ligand (L1) with 1a-Ms or Ligand (L2) with 1a to do the computational studies of the regioselectivity determining step. Because the reaction probably did not yield the products (<3%) under PtBu₂Me (L2) with substrate 1a-Ms, which are used for DFT calculation (Figure 3). It's hard for readers to trust the mechanism of a non-working reaction even under reasonable analysis of the regioselectivity.

Please see the table on page S17 of the SI, the average yield is 93.8% under L1 with 1a and only 10.8% under L2 with 1a. On page S25 of the SI, the average yield of 3a-Ms is 32% under the optimized condition (L1 and 1a-Ms). It looks hard to generate the product (3a-Ms) under the weaker L2 with 1a-Ms.

Reviewer 4

After reading the manuscript and the responses of the authors, I think the authors still need to use Ligand (L1) with 1a-Ms or Lignad (L2) with 1a to do the computational studies of the regioselectivity determining step. Because the reaction probably did not yield the products (<3%) under PtBu2Me (L2) with substrate 1a-Ms, which are used for DFT calculation (Figure 3). It's hard for readers to trust the mechanism of a non-working reaction even under reasonable analysis of the regioselectivity. Please see the table on page S17 of the SI, the average yield is 93.8% under L1 with 1a and only 10.8% under L2 with 1a. On page S25 of the SI, the average yield of 3a-Ms is 32% under the optimized condition (L1 and 1a-Ms). It looks hard to generate the product (3a-Ms) under the weaker L2 with 1a-Ms.

Thank you for your comments. We chose to use **1a-Ms** with isoprene (**2b**) and **L2** as models for our DFT calculations as our model reaction – with **1a-Ts**, myrcene (**2a**) and **L1** – was sufficiently computationally demanding that attempted transition state calculations failed to converge, despite multiple attempts and even at a lower level of theory; this was also the case when we tried to perform calculations with **1a-Ms** and **L1**. When we used **1a-Ts** and **L2**, it was also challenging to get the transition state calculations to converge, and when it did, the added benzene ring introduced additional dispersion that led to errors (DFT is known to struggle with dispersion). Parallel with these efforts, we followed up on experimentally validating the model substrates/ligand we chose for our calculations, as you rightly point out that this was lacking. Indeed, under our standard reaction conditions (1 mol% Pd₂(dba)₃ + 4 mol% ligand, 100 °C), the reactivity with **1a-Ms**, **2b**, and **L2** was too poor to measure regioselectivity, as you predicted. However, when we increased the catalyst loading to 2 mol% Pd₂(dba)₃ + 8 mol% ligand and increased the temperature to 120 °C, we were able to isolate product **3a-Ms** in 34% yield and >95:5 regioselectivity, favoring the 3-substituted product. We realized that an important detail was pre-stirring our ligand and Pd source, since we were using the tetrafluoroborate salt of **L2**; no reactivity was observed without that pre-stir. This reaction was duplicated and the results and experimental details have been added to the SI (S35). While this is a slight variation from our standard conditions, we contend that these minor changes (temperature and catalyst loading) should not have any impact on the structure of the relevant transition state in the regioselectivity-determining step, supporting the validity of our DFT model. We have also added language to the text to indicate this experimental corroboration of our DFT findings:

“While this combination was suboptimal under our standard catalytic conditions, with a slight increase in catalyst loading and reaction temperature, we were able to isolate product **3ab-Ms** in 34% yield and >95:5 r.r.; this 3-selectivity is comparable to our experimental model reaction under nearly identical conditions.”